# Energy, Exergy, and Economic Analysis of Cryogenic Distillation and Chemical Scrubbing for Biogas Upgrading and Hydrogen Production

**Esfandiyar Naeiji [1], Alireza Noorpoor [1,*] and Hossein Ghanavati [2]**

[1] School of Environment, College of Engineering, University of Tehran, Tehran 1417853111, Iran; es_naeiji@ut.ac.ir

[2] Department of Microbial Biotechnology, Agricultural Biotechnology Research Institute of Iran (ABRII), Agricultural Research, Extension, and Education Organization (AREEO), Karaj 3135933151, Iran; ghanavatih@abrii.ac.ir

[*] Correspondence: noorpoor@ut.ac.ir

**Abstract:** Biogas is one of the most important sources of renewable energy and hydrogen production, which needs upgrading to be functional. In this study, two methods of biogas upgrading from organic parts of municipal waste were investigated. For biogas upgrading, this article used a 3E analysis and simulated cryogenic separation and chemical scrubbing. The primary goal was to compare thermoeconomic indices and create hydrogen by reforming biomethane. The exergy analysis revealed that the compressor of the refrigerant and recovery column of MEA contributed the most exergy loss in the cryogenic separation and chemical scrubbing. The total exergy efficiency of cryogenic separation and chemical scrubbing was 85% and 84%. The energy analysis revealed a 2.07% lower energy efficiency for chemical scrubbing. The capital, energy, and total annual costs of chemical absorption were 56.51, 26.33, and 54.44 percent lower than those of cryogenic separation, respectively, indicating that this technology is more economically feasible. Moreover, because the thermodynamic efficiencies of the two methods were comparable, the chemical absorption method was adopted for hydrogen production. The biomethane steam reforming was simulated, and the results indicated that this method required an energy consumption of 90.48 $\frac{MJ}{kg_{H_2}}$. The hydrogen production intensity equaled 1.98 $\frac{kmole_{H_2}}{kmole_{biogas}}$ via a 79.92% methane conversion.

**Keywords:** cryogenic separation; exergy; biogas upgrading; hydrogen; chemical absorption

## 1. Introduction

With increased awareness of the consequences of global warming and the recent increase in fuel prices, several efforts were made to develop studies on sustainable energy. To mitigate greenhouse gas emissions, researchers have emphasized the importance of using renewable resources to offset the consumption of fossil fuels. Presently, the $CO_2$ level in the atmosphere is approximately 100 ppm higher than it was during the pre-industrial era (280 ppm). The Kyoto Protocol ordered the European Union and several of the world's 37 industrialized countries to reduce greenhouse gas emissions by 5.2 percent in order to counteract global warming. Besides, the Copenhagen Climate Change Accord aimed to limit global warming up to 2 °C by 2100 [1].

Biogas is a dependable renewable energy source typically generated via the anaerobic digestion of biomass. Globally, the potential for biogas production from available substrates is now between 10,100 and 14,000 TWh. If fully utilized, the generated energy can offset between 6 and 9% of the world's initial energy consumption and serve as a viable alternative to fossil fuels [2].

Carbon dioxide is the most significant man made greenhouse gas contributing to global warming. Several temperature lowering technologies must be used to achieve

the Paris Climate Accords' goal of a 2% decrease in global temperature. Biogas plays an essential role in the energy market as a renewable energy source. To boost thermal value, it is important to eliminate carbon dioxide and upgrade to a higher fuel standard. With a rise in biogas output and a high carbon dioxide concentration, biogas upgrading, $CO_2$ usage, and $CO_2$ absorption have all gotten a lot of attention. Because raw biogas should be compressed under high pressure and external cooling is required in some systems, this process requires a significant amount of energy. As a result, optimizing and comparing various refrigeration systems is critical. Furthermore, deviations in the thermodynamic properties of $CH_4/CO_2$ mixtures can significantly impact the design and performance of the system, necessitating a more thorough understanding of the properties [3].

CH$_4$ and carbon dioxide are the significant constituents of biogas, and the ratio of methane to carbon dioxide in the extracted biogas from fertilizers or sewage sludge is about 60 to 40. The proportions of these two components change depending on the kind of feedstock and the bioreactor's operating parameters. Municipal solid waste, as well as animal waste and agricultural waste, may all be used to make biogas [4]. Adsorption (for example, Pressure Swing Adsorption), chemical scrubbing, water scrubbing, organic solvent scrubbing, membrane separation, and cryogenic separation processes can be used to upgrade raw biogas [5–8]. The appropriate technology for raw biogas upgrading is determined by the end-use of biogas, the process's cost-effectiveness, and efficiency [9]. Mehrpooya et al., simulated a cryogenic biogas upgrading process [10]. The upgraded biogas can be used to generate electricity, heat, or a combination of the two, CNG, or as a raw material for hydrogen production [7,11–13]. Steam reforming is one of the most established methods to produce hydrogen [14,15]. Hashemiet al., simulated the process of cryogenic distillation and chemical absorption for biogas upgrading and examined it in terms of energy consumption [16]. Aspen HYSYS software was used to evaluate the two processes, and the simulation results revealed that the cryogenic distillation method is more energy-efficient than chemical absorption. Yousef et al., modeled a low-temperature biogas upgrading process. The simulation was performed using the Aspen HYSYS software. In the separation stage, two columns were used, and in the pressure increase stage, four stages were simulated [17]. In this simulation, raw biogas contained 40% carbon dioxide, and the purity of the product carbon dioxide was 99.9%. Vilardi et al., using the Aspen PLUS software, simulated three biogas upgrading processes, including water scrubbing, chemical scrubbing, and membrane separation. They showed that rinsing with water has the highest exergy efficiency and that membrane separation has the lowest exergy efficiency [18].

There are various methods to produce hydrogen, one of the most important of which is methane steam reforming [19]. This process includes reforming, water–gas shift reactors, and separation equipment.

Biogas upgrading processes have great importance in energy or hydrogen extraction from biomass through anaerobic digestion. In previous research, the energy and process parameters of these methods were compared. This study aims to comparatively evaluate the energy, exergy, and economic parameters of the two main biogas upgrading processes, i.e., a process with cryogenic distillation configuration and chemical scrubbing, and to investigate the performance, yield, and cost of the process equipment. The combination of economic evaluation and energy and exergy assessment result in a better selection of the process. The high purity of the resultant methane from the two processes is one of the reasons for choosing these two processes, which enables hydrogen production from the product of the reforming process. The assessment and comparison of the thermodynamic and economic aspects of these two processes allow for the selection of the best process, considering different parameters to reach the optimum design of the energy production cycle from biomass (municipal waste). Furthermore, the assessment of the different stages of the process and equipment can be performed to determine the major energy-consuming equipment exergy loss and the cost of these processes. The present study simulated and investigated the two cryogenic absorption and chemical scrubbing methods in thermoeconomic terms using the MEA solvent to determine which methods would be suitable for

hydrogen production via steam reforming the produced biomethane. After comparing economic and technical results and introducing the desirable alternative, this paper conducted comprehensive energy, exergy, and economic studies (3E analysis) and simulated a hydrogen production process. The current study used a 3E analysis to simulate an ideal biomethane hydrogen production process, as well as define and compute competitive factors, including energy consumption and hydrogen production intensities. The biogas characteristics in this study were obtained from laboratory digestion results for an organic fraction of municipal waste with alkaline, thermal, and $H_2O_2$ pretreatment.

## 2. Material and Methods

Figure 1 summarizes the research steps. First, the process is simulated, and energy, economic, and exergy analyses are performed. Then, the optimal state results are determined, and the hydrogen production process is simulated. The relationships required for these simulations are provided below.

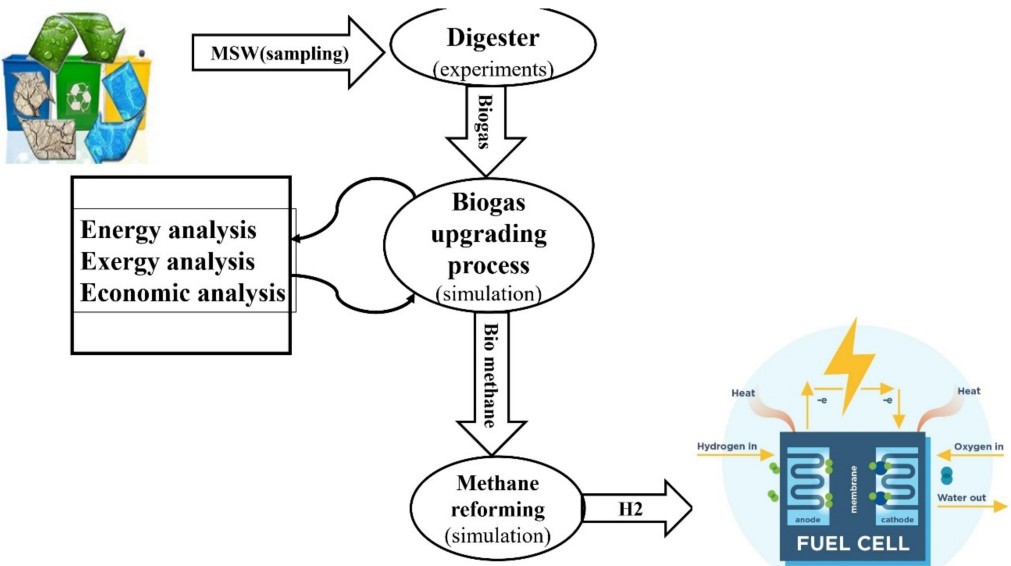

**Figure 1.** Summary of research steps.

### 2.1. Cryogenic Separation Process of Biogas

The diagram in Figure 2 indicates the cryogenic separation of biogas into methane and carbon dioxide. First, biogas, composed of 62% methane and 38% carbon dioxide, was compressed in a 3-phase compressor operating at a pressure of 105–4625 kPa. After passing through the E-102 and E-104 heat exchangers, it enters the distillation column at a temperature of −70 °C. A compressor (K-103), a heat exchanger (E-105), and a Joule–Thomson valve make up the refrigeration cycle, which is meant to keep the biogas at the requisite cryogenic temperature.

The low temperature of the biomethane flow was used to cool the compressed biogas in the E-102 heat exchanger, requiring less energy to reduce the biogas temperature in the cryogenic phase. A ten-tray partial condenser was used in the distillation column for biogas separation. The liquid and steam products of the condenser were transported to an E-102 heat exchanger and combined in a mixer, with the biomethane flow as the heat exchanger's output. The refrigeration cycle in this process was simulated using R-170 85%–R290 15% operating fluid. Other fluids (R-170, and R170 93%–R600 7%) were simulated for the refrigeration cycle, and fluid R-170 85%–R290 15% had the lowest energy consumption and cost. The refrigerant temperature was increased to 3905 kPa by K-103 compressor, and the refrigerant was cooled down to 30 °C in a condenser before Joule–Thompson expansion. To this end, the condenser utilized water. It is worth noting that cooling with cold water was employed in the biogas

compression phase. The cold-water temperatures were assumed to be 20 °C and 25 °C in the inlet and outlet, respectively.

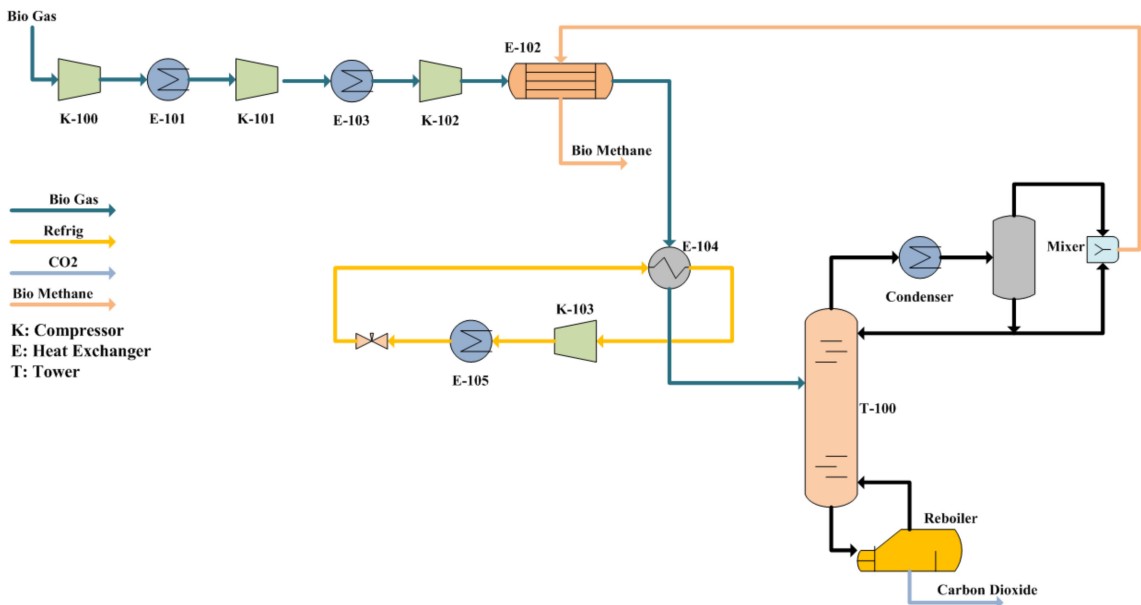

**Figure 2.** Schematic view of the biogas cryogenic distillation process.

### 2.2. Biogas Separation Process via the Chemical Absorption Method

The chemical absorption of carbon dioxide from biogas is seen in Figure 3. Two absorption and desorption columns were used in the procedure. The solvent and biogas were introduced to the absorption column from the top and bottom, respectively, and gas absorption was achieved by a chemical reaction between the solvent and carbon dioxide. Thus, purified gas was produced at the absorption column's apex. Regarding the presence of water and the solvent concentrations in the gas flow, a cooler at the top of the absorption column was used to reduce the gas temperature. A two-phase separator separated the liquids, primarily composed of water and solvent. As a result, the solvent loss caused by the biomethane gas flow was minimized.

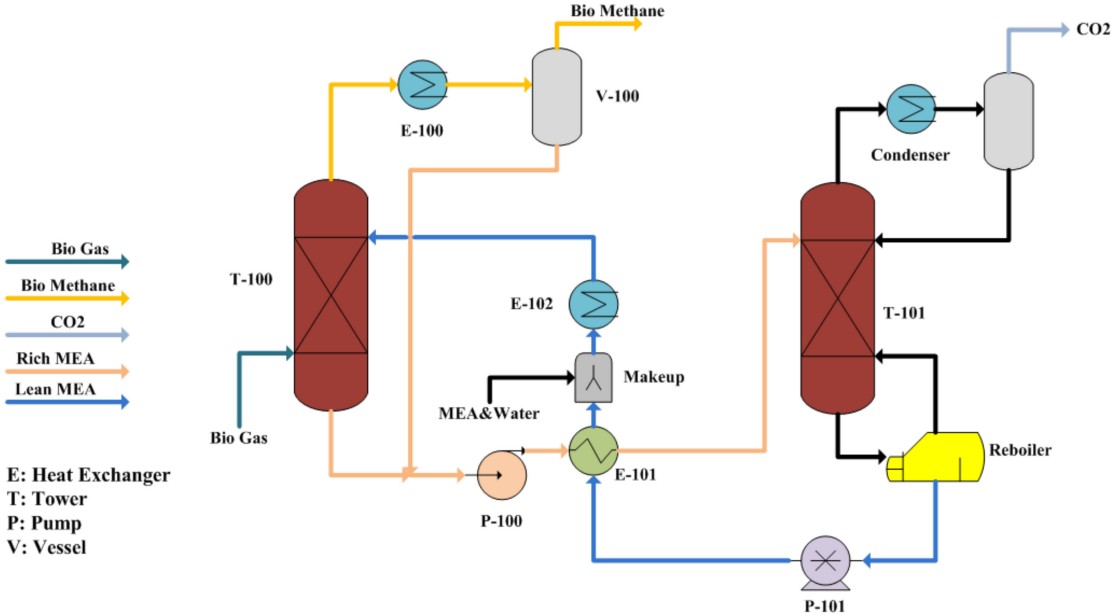

**Figure 3.** Schematic view of the chemical absorption process.

The absorption and desorption columns were packed similarly to the work of Vilardi et al. [18]. The packing pall rings were made of metal with a 16 mm diameter. The columns were 10 m tall, and a one-meter separation between each balance phase was assumed. The column diameter was determined using the Aspen HYSYS V.12 software. The diameters of the absorption and desorption columns were calculated to be 0.5282 m and 0.4625 m, respectively. The lean MEA solvent was loaded at a rate of 0.25–0.3 carbon dioxide mol per mole of MEA [20]. The flow rate of the solvent was determined using the standard value of carbon dioxide in the produced biomethane. The maximum amount of carbon dioxide for biomethane that could be transferred to the gas pump was 4% V [18]. Thus, during the simulation process, the solvent flow rate was determined based on the loading, and the permissible value of carbon dioxide was estimated to be 300 kmol/h.

The stripper column's operating pressure (solvent recovery) was between 150 and 300 kPa, and the MEA solvent, with a temperature of 117–120 °C, exited the reboiler and cooled to a certain extent after exchanging heat with the rich MEA. Makeup streams of solvent and water were injected into the lean MEA produced by the reboiler to compensate for the solvent and water losses during the solvent recovery phase. In this case, the computations were performed using the Makeup logic in the Aspen HYSYS software, and the makeup water flow rate was estimated to be 5 kmol/h. The loss rate of the MEA solvent was zeroed due to the placement of a cooler and separator to the top of the absorption column; as a result, the Makeup logic was not regarded as a value for the solvent. Moreover, the condenser temperature was set to 45 °C to minimize the amount of water in the carbon dioxide produced at the top of the desorption column. Based on the molar ratio of the lean solvent and the loading rate calculated by Recycle logic ($0.2486 \frac{\text{mole CO2}}{\text{mole MEA}}$), the optimal value of the return ratio in the desorption column was determined to be 0.48.

### 2.3. Fluid Package for the Carbon Dioxide–Methane System

Xiong et al. [21] used Peng–Robinson's equation of state for the cryogenic separation process. Li et al. [22] conducted an exhaustive study of the phase thermodynamics of the cryogenic separation of carbon dioxide and methane. Tan et al. [3] investigated a number of thermodynamic equations that might be applied in the cryogenic separation of biogas. Their study introduced multiple equations, including SRK (Redlich–Kwong-Soave) Peng–Robinson, and RK. Likewise, Li et al. [23] conducted a comprehensive study in which they analyzed several equations, including SRK, RK, PR, and Patel–Teja, to predict the carbon's steam–liquid balance dioxide–methane system and discovered that Peng–Robinson's equation of state possessed the highest accuracy.

Yousef et al. [24] described a novel method for purifying biogas (carbon-dioxide–Biomethane separation). They stated that Peng–Robinson's equation of state was highly accurate and was well suited for simulating the cryogenic distillation of biogas. Yousef et al. [25] used Peng-fluid Robinson's package in conjunction with the Aspen HYSYS software to calculate the thermodynamic characteristics of fluids throughout the process in a similar investigation. The acid–gas–chemical solvents equation of state, which is available in the Aspen properties data bank, is utilized in the chemical scrubbing process. This equation of state includes the relationships associated with the MEW process equipment [26].

In this simulation Peng–Robinson's equation of state was used in the cryogenic separation phase, and the Acid–Gas–Chemical Solvents package was employed to simulate the chemical absorption process.

Peng–Robinson's equation of state [27].

$$E_k = \frac{RT}{v - b} - \frac{a\alpha}{v^2 + 2bv - b^2} \tag{1}$$

$$a = \frac{0.45724 \, R^2 T_c^2}{P_c} \tag{2}$$

$$b = \frac{0.07780 \, RT_c}{P_c} \tag{3}$$

$$\alpha = \left(1 + k\left(1 - T_r^{0.5}\right)\right)^2 \tag{4}$$

$$k = 0.37464 + 1.54226\,\omega + 0.26992\omega^2 \tag{5}$$

### 2.4. Solvent Selection for the Chemical Absorption Process

Numerous solvents are available to remove carbon dioxide and hydrogen sulfide, including diethanolamine, diglycolamine, a piperazine and methyl diethanolamine mix, and monoethanolamine. The present study employed a biogas mixture similar to that used by Vilardi et al. [18] and a monoethanolamine solvent To simulate the chemical absorption of carbon dioxide. This solvent is cost-effective, has a low molecular weight, and has little methane hydrocarbon solubility. As a result, methane leakage is reduced, and the solvent has a high capacity for carbon dioxide absorption [28]. MEA is typically dissolved in water at a concentration of 12–30 wt% [29]. Due to the high carbon dioxide concentration in biogas (32% mol), a 30 wt% MEA was used in this simulation—the following equilibrium reactions occurred in this process: [20]:

$$2H_2O \boxed{\leftrightarrow} H_2O^+ + OH^- \tag{6}$$

$$2H_2O + CO_2 \boxed{\leftrightarrow} HCO_3^- + H_3O^+ \tag{7}$$

$$H_2O + HCO_3^- \boxed{\leftrightarrow} CO_3^{2-} + H_3O^+ \tag{8}$$

$$MEAT^+ + H_2O \boxed{\leftrightarrow} MEA + H_3O^+ \tag{9}$$

$$InK_j = A_j + \frac{B_j}{T} + C_j InT + D_j T \tag{10}$$

$$MEA + H_2O + CO_2 \boxed{\leftrightarrow} MEACOO^- + H_3O^+ \tag{11}$$

$$OH^- + CO_2 \boxed{\leftrightarrow} HCO_3^- \tag{12}$$

### 2.5. Energy Analysis

This research presents two biogas-upgrading structures. The first structure used traditional cryogenic separation, while the second used chemical absorption to improve biogas. As a consequence, each structure's thermodynamics may differ. The energy efficiency of the cryogenic separation process ($\eta_{energy}^{cryo}$) and the energy efficiency of the chemical absorption method of biogas separation ($\eta_{energy}^{chem}$) are denoted in this research as Equations (13) and (14), respectively.

$$0\eta_{energy}^{cryo} = \frac{\dot{m}_{biomethane}\,LHV_{biomethane}}{\dot{m}_{biogas}\,LHV_{biogas} + \dot{W}_{comp} + \dot{Q}_{Reb}} \tag{13}$$

$$\eta_{energy}^{chem} = \frac{\dot{m}_{biomethane}\,LHV_{biomethane}}{\dot{m}_{biogas}\,LHV_{biogas} + \dot{W}_{pump} + \dot{Q}_{Reb}} \tag{14}$$

where $\dot{m}_{biomethane}$ and $\dot{m}_{biogas}$ denote the input biomethane and biogas mass flow rate in kg/h.

### 2.6. Economic Analysis

As previously stated, this study examined and simulated two distinct biogas upgrading processes. Total Annual Cost (TAC), which is a function of the payback period, capital cost, and energy cost parameters, which has been used in this section as a benchmark for both processes. The TAC formulation is based on Equation (10) [30]:

$$TAC = \frac{Capital\ Cost}{payback\ period} + Energy\ Cost \tag{15}$$

The capital cost is the sum of direct costs, indirect costs, startup costs, working capital, and allowance funds used during construction. Direct and indirect costs were defined as fixed costs. In Equation (4), the payback period was considered to be three years [30].

### 2.7. Exergy Analysis

Exergy is a vastly important and complex term whose consideration, application in any industry, particularly petroleum, gas, petrochemistry, and analysis, enables us to scientifically define the term and identify various process points that result in exergy loss and reduction. Moreover, less energy consumption and energy loss can increase and optimize production. The exergy of a thermodynamic system is the amount of practical work that that system can accomplish. Furthermore, the exergy of the two cryogenic separation and chemical scrubbing procedures for biogas was investigated in this study. For this, many concepts were estimated, including exergy loss and efficiency, and the results were compared. The exergy of a flow ($\dot{E}_i$) is computable based on Equation (16):

$$\dot{E}_i = e_i \times \dot{m}_i t \tag{16}$$

where $e_i$ is the specific exergy in $\frac{kj}{kg}$, and $\dot{m}_i$ is the flow rate i in $\frac{kg}{h}$.

Exergy: Each current has four different components, including kinetic $E_k$, potential $E_p$, physical $E_{ph}$, and chemical $E_{ch}$ exergy. Equations for calculating exergy are found in Equations (17) to (20).

$$E_k = \frac{m \times v^2}{2} \tag{17}$$

$$E_p = m \times g \times (z - z_o) \tag{18}$$

$$E_{ph} = \Delta h - T_o \Delta s \tag{19}$$

$$E_{ch} = \sum (\mu_o - \mu_{oo}) \tag{20}$$

The total energy of a flow is only included in the exergy analysis of the cryogenic separation process because there is no chemical reaction, and separation is solely based on the boiling point. Both methodologies have failed to account for potential and kinetic exergies. In the chemical scrubbing approach, the total exergy of each flow comprises both physical and chemical exergies produced by the interaction between the solvent and carbon dioxide gas. Thus, for cryogenic separation and chemical scrubbing, the total exergy of each flow can be calculated using Equations (21) and (22):

$$E_i^{Total,\ Cryo} = \dot{E}_i^{PH} \tag{21}$$

$$E_i^{Total,\ Chem} = \dot{E}_i^{PH} + \dot{E}_i^{CH} \tag{22}$$

Exergy loss for equipment ith is calculated by Equation (23):

$$E_i^{loss} = \dot{E}_{in} - \dot{E}_{out} \tag{23}$$

The input and output exergy for the equipment used in the cryogenic separation and chemical scrubbing of biogas with the MEA solvent are shown in Table 1. Similarly, Equations (24) and (25) present the rate of total exergy efficiency for each the cryogenic separation and chemical scrubbing process:

$$\eta_{Exergy}^{Cryo} = \frac{\sum \dot{E}_{out}^{cryo}}{\sum \dot{E}_{in}^{cryo}} \tag{24}$$

$$\eta_{Exergy}^{Chem} = \frac{\sum \dot{E}_{out}^{Chem}}{\sum \dot{E}_{in}^{Chem}} \tag{25}$$

where $\dot{E}_{Biomethane}$ and $\dot{E}_{Bio\,Gas}$ denote the total exergy of biomethane and biogas flow, respectively, and $\dot{W}_{Pump}$ and $\dot{W}_{Comp}$ are the consumed power in the pump and compressors, respectively. $\dot{E}_{Reb}$ represents the reboiler exergy in the biogas cryogenic separation columns and solvent recycling in the chemical scrubbing system. Equations (26) and (27) are used for its estimation:

$$\dot{E}_{Reb}^{Cryo} = \dot{E}_{BFW} - \dot{E}_{condensate} \tag{26}$$

$$\dot{E}_{Reb}^{Chem} = \dot{E}_{LPS} - \dot{E}_{LPC} \tag{27}$$

**Table 1.** Exergy relations and equations for main process equipment application [31–41].

| Component | $\dot{E}_{in}(KW)$ | $\dot{E}_{out}(KW)$ | Exergy Efficiency |
|---|---|---|---|
| Compressor | $\dot{E}_{inlet} + \dot{W}$ | $\dot{E}_{outlet}$ | $\eta_{ex} = \frac{\Sigma(\dot{m}.e)_i - \Sigma(\dot{m}.e)_o}{W}$ |
| Heat Exchanger | $\dot{E}_{inlet\,Hot} + \dot{E}_{inlet\,Cold}$ | $\dot{E}_{outlet\,Hot} + \dot{E}_{outlet\,Cold}$ | $\eta_{ex} = 1 - \left[\left\{\frac{\Sigma_1^n \dot{m}\Delta e}{\Sigma_1^n \dot{m}\Delta h}\right\}_h - \left\{\frac{\Sigma_1^m \dot{m}\Delta e}{\Sigma_1^m \dot{m}\Delta h}\right\}_e\right]$ |
| Column | $\dot{E}_{Feed} + \dot{E}_{Reb}$ | $\Sigma \dot{E}_{Prod} + \dot{E}_{Cond}$ | |
| Expansion valves | $\dot{E}_{inlet}$ <br> $E^{Ph} = E^{\Delta T} + E^{\Delta P},$ <br> $E^{\Delta T} = \int_T^{T_o} \frac{T-T_o}{T} dh,$ | $\Sigma \dot{E}_{outlet}$ | $\eta_{ex} = \frac{e_o^{\Delta T} - e_i^{\Delta T}}{e_i^{\Delta T} - e_o^{\Delta T}}$ |
| Pump | $\dot{E}_{inlet} + \dot{W}$ | $\dot{E}_{outlet}$ | $\eta_{ex} = \frac{\Sigma(\dot{m}.e)_i - \Sigma(\dot{m}.e)_o}{W}$ |

## 3. Results

### 3.1. Simulation

The Aspen HYSYS software was used in this study to simulate the cryogenic separation and chemical absorption of biogas. Figures 4 and 5 show the simulated process. In both processes, the height of the columns is the same, and both are considered to have 10 steps.

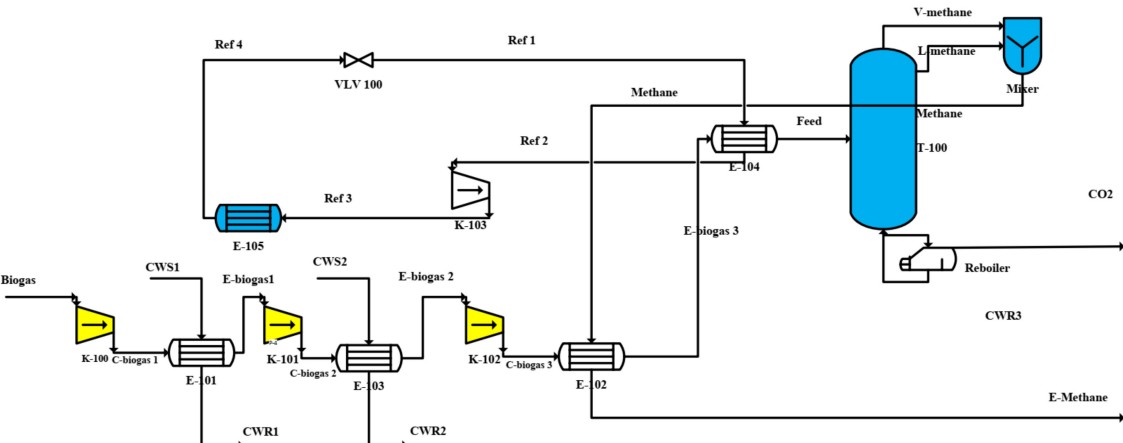

**Figure 4.** Schematic view of the biogas cryogenic separation simulation.

The simulator is run in static mode after setting the feed stream and equipment characteristics, such as the reflux ratio, number of columns steps, establishing the ideal condition of the refrigeration cycle, and so on. Based on the mass balance and energy balance, simulation results are obtained. The operating conditions of each of the streams obtained from the simulation results for each process are presented in Tables 2 and 3. As can be seen, the purity of the output methane is 0.9874 and 0.9873 for each of the cryogenic separation and chemical separation processes, respectively.

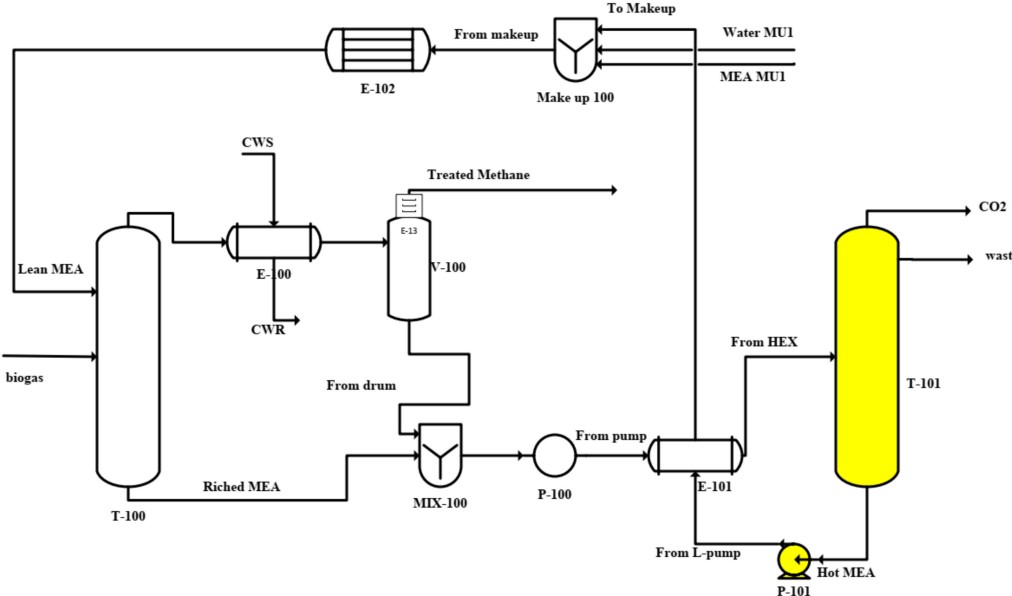

**Figure 5.** Schematic view of the biogas separation simulation via the chemical absorption method.

**Table 2.** Simulation results for operating conditions and composition of flows in the cryogenic distillation process.

| Stream | T (°C) | P (kPa) | F ($\frac{kmole}{h}$) | % CH$_4$ | % CO$_2$ | % R- | % R- | % H$_2$O$_2$ |
|---|---|---|---|---|---|---|---|---|
| Biogas | 30 | 105 | 25.72 | 0.62 | 0.38 | 0 | 0 | 0 |
| E-biogas 1 | 30 | 380 | 25.72 | 0.62 | 0.38 | 0 | 0 | 0 |
| E-biogas 2 | 30 | 1350 | 25.72 | 0.62 | 0.38 | 0 | 0 | 0 |
| E-biogas 3 | 4.308 | 4585 | 25.72 | 0.62 | 0.38 | 0 | 0 | 0 |
| Methane | −90.3 | 3500 | 16.15 | 0.9874 | 0.0126 | 0 | 0 | 0 |
| CO2 | 5.577 | 4000 | 9.572 | 0.62 | 0.38 | 0 | 0 | 0 |
| V-methane | −90.3 | 3500 | 14.9 | 0.9888 | 0.0112 | 0 | 0 | 0 |
| Ref3 | 136.8 | 3904 | 50.54 | 0.9874 | 0.0126 | 0 | 0 | 0 |
| C-biogas 1 | 156.4 | 380 | 25.72 | 0.62 | 0.38 | 0 | 0 | 0 |
| C-biogas 2 | 155 | 1350 | 25.72 | 0.62 | 0.38 | 0 | 0 | 0 |
| C-biogas 3 | 152.7 | 4625 | 25.72 | 0.62 | 0.38 | 0 | 0 | 0 |
| E-methane | 142.7 | 3460 | 16.15 | 0.9874 | 0.0126 | 0 | 0 | 0 |
| Feed | −70 | 4500 | 25.72 | 0.62 | 0.38 | 0 | 0 | 0 |
| L-methane | −90.3 | 3500 | 1.25 | 0.9714 | 0.0286 | 0 | 0 | 0 |
| Ref4 | 30 | 3904 | 50.54 | 0 | 0 | 0.85 | 0.15 | 0 |
| CWSs&CWRs | | | | 0 | 0 | 0 | 0 | 1 |

**Table 3.** Simulation results for operating conditions and composition of flows in the chemical process.

| Stream | T (°C) | P (kPa) | F ($\frac{kmole}{h}$) | %CH$_4$ | % CO$_2$ | % MEA | % H$_2$O |
|---|---|---|---|---|---|---|---|
| Biogas | 30 | 105 | 25.72 | 0.62 | 0.38 | 0 | 0 |
| From MEA Cooler | 30 | 110 | 300 | 0 | 0.0291 | 0.1169 | 0.854 |
| CWS2 | 15 | 500 | 2557 | 0 | 0 | 0 | 1 |
| From L-Pump | 117.5 | 250 | 295 | 0 | 0.0296 | 0.1189 | 0.8515 |
| To Makeup | 66.19 | 195 | 295 | 0 | 0.0296 | 0.1189 | 0.8515 |
| To Cooler | 33.05 | 105 | 17.13 | 0.9309 | 0.0265 | 0 | 0.0426 |
| To Drum | 20 | 105 | 17.13 | 0.9309 | 0.0265 | 0 | 0.0426 |
| Treated Methane | 12 | 104 | 16.15 | 0.9873 | 0.0127 | 0 | 0 |
| To Pump | 60.53 | 105 | 309 | 0 | 0.0584 | 0.1135 | 0.8280 |
| Hot MEA | 117.5 | 180 | 295 | 0 | 0.0296 | 0.1189 | 0.8515 |
| LPC | 138.4 | 344.7 | 80 | 0 | 0 | 0 | 1 |
| CO2 | 45 | 180 | 9.855 | 0.0005 | 0.9454 | 0 | 0.0541 |
| Lean MEA | 30 | 110 | 300 | 0 | 0.0291 | 0.1169 | 0.854 |
| From Makeup | 66.22 | 195 | 300 | 0 | 0.0291 | 0.1169 | 0.854 |
| CWR2 | 20 | 500 | 2557 | 0 | 0 | 0 | 1 |
| From HEX | 103 | 200 | 309 | 0 | 0.0584 | 0.1135 | 0.8280 |
| CWS | 15 | 500 | 22.90 | 0 | 0 | 0 | 1 |
| CWR | 30 | 500 | 22.90 | 0 | 0 | 0 | 1 |
| From Drum | 20 | 105 | 0.3552 | 0 | 0.0008 | 0.0008 | 0.9984 |
| Rich MEA | 60.58 | 105 | 308.6 | 0 | 0.0585 | 0.1137 | 0.8279 |
| From Pump | 60.57 | 300 | 309 | 0 | 0.0584 | 0.1135 | 0.8280 |
| LPS | 138.4 | 344.7 | 80 | 0 | 0 | 0 | 1 |
| Waste | 45 | 180 | 4.093 | 0 | 0.0015 | 0.0009 | 0.9977 |

The changes in the concentration of methane and carbon dioxide (in the gas phase) and the changes in temperature and pressure in the cryogenic distillation column and the chemical adsorption column are shown in Figures 6 and 7. Number 1 is for the upper stage, and number 10 is for the lower stage (excluding reboilers and condensers).

As shown in Figure 6, the pressure and temperature increase as the gas flows from the top to the bottom of the column. The cooled biogas enters the column from the middle or from the 5th stage. We expect methane to have the lowest boiling point and carbon dioxide to have the highest at the bottom of the tower because methane has the lowest boiling point. The temperature is lower near the top of the tower, and the gas flow is high in methane. Figure 6a shows that the temperature is around −40 °C. While the temperature trend is upward, because the middle of the column is where the cold feed (−70 °C) enters the tower, it makes the temperature increase trend less horizontal from the bottom to the top of the tower in the vicinity of the entry point.

Figure 6c,d show that the bottom of the column (step 10) is the hottest point of the column (near the reboiler), this point is rich in carbon dioxide, and the top of the tower is the coldest point of the tower (near the condenser) and is rich in methane.

Figure 7 shows the temperature, pressure, and concentration of compounds in the gas phase in different parts of column. Biogas enters from the bottom and the MEA solution from the top of the packed column. The MEA solvent does not dissolve methane. Therefore, methane concentration is expected to increase from the bottom to the top of the column. However, as shown in Figure 7a–c, this is because the temperature in the middle of the column is higher and the concentration of water vapor (and the total flow rate of the gas phase) is higher. As a result, despite the fact that the quantity of methane remains constant while the amount of carbon dioxide drops, the methane concentration initially declines and subsequently increases, as shown in Figure 7d. As the contact of carbon dioxide with the solvent increases along the column, the concentration of carbon dioxide decreases.

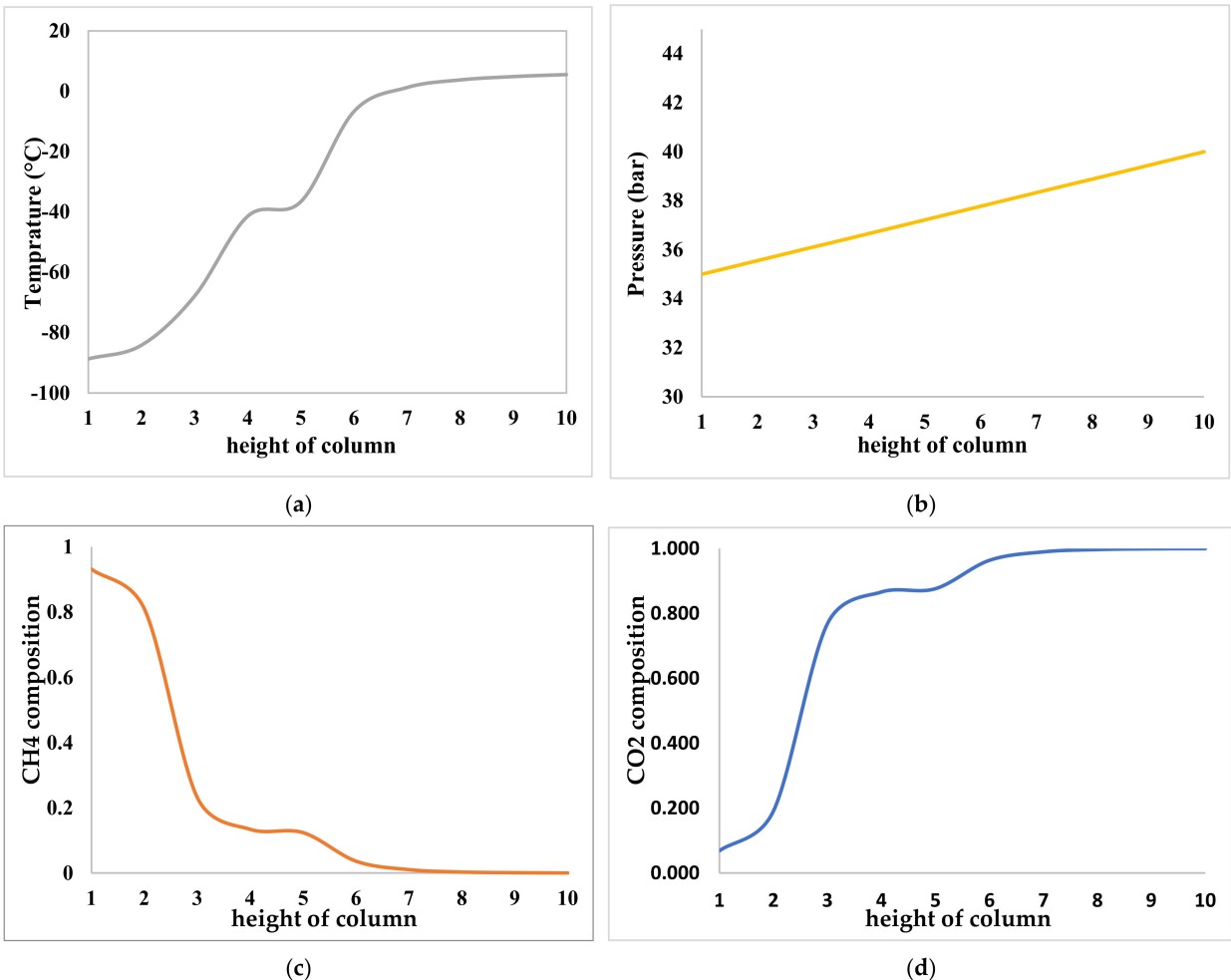

**Figure 6.** Temperature (**a**), pressure (**b**), $CH_4$ composition, (**c**) and $CO_2$ composition (**d**) in the cryogenic distillation column.

Regarding the validity of the results, it can be seen that previous research predicted about 99% separation for methane in the process of cryogenic biogas separation at −70 °C [42,43]. Previous articles have reported a range from 97 to 99% for the purity of methane in the chemical scrubbing process [6,43,44].

### 3.2. Sensitivity Analysis

The analysis of the effect of the reflux ratio of cryogenic distillation separation columns and chemical scrubbing on the boiler duty is presented in Figure 8. Figure 8 shows that the effect of the reflux ratio on the duty of the separation column boiler is much more noticeable in the chemical adsorption method. The chemical scrubbing duty of boilers is also greater than the cryogenic separation duty. The boiling point of the separating chemicals is substantially higher in the chemical adsorption approach than in the cryogenic separation method owing to the nature and kind of separation. In the cryogenic separation reboiler, only methane and carbon dioxide are evaporated, while in the chemical adsorption process reboiler, the water evaporates, which has a higher boiling point and heat capacity. Therefore, to evaporate the compound, a higher boiling point is required, and more energy is consumed in the reboiler. In the chemical scrubbing process, the duty of the reboiler was ascending to a reflux ratio of 1.9 and was constant from this value onwards because all carbon dioxide is excreted from the input solution and is obtained in the condenser.

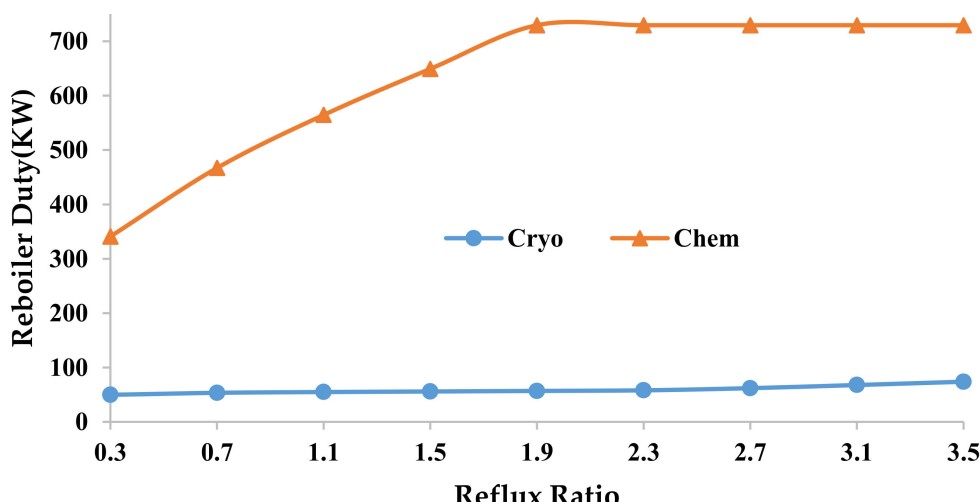

**Figure 7.** Temperature (**a**), pressure (**b**), CH$_4$ composition, (**c**) and CO$_2$ composition (**d**) in the chemical absorption column.

**Figure 8.** Effect of the reflux ratio on the duty of the reboiler in the separation columns.

The effect of the changes in the biogas flow rate on the purity of output methane, as a factor to compare process compatibility, is presented in Figure 9. Figure 9 shows that each of the processes are adaptable to some extent against the biogas flow rate without changing the main characteristics. However, in the cryogenic distillation process, fewer changes in

the purity of the output methane were observed against changes in the biogas flow rate. With a high flow rate (+5%) in both processes, a sharp decrease in biogas purity is observed in terms of deviation from the design characteristics. In the process of chemical separation, by reducing the biogas flow rate, if other characteristics were constant the output methane concentration will be more than 99.9%.

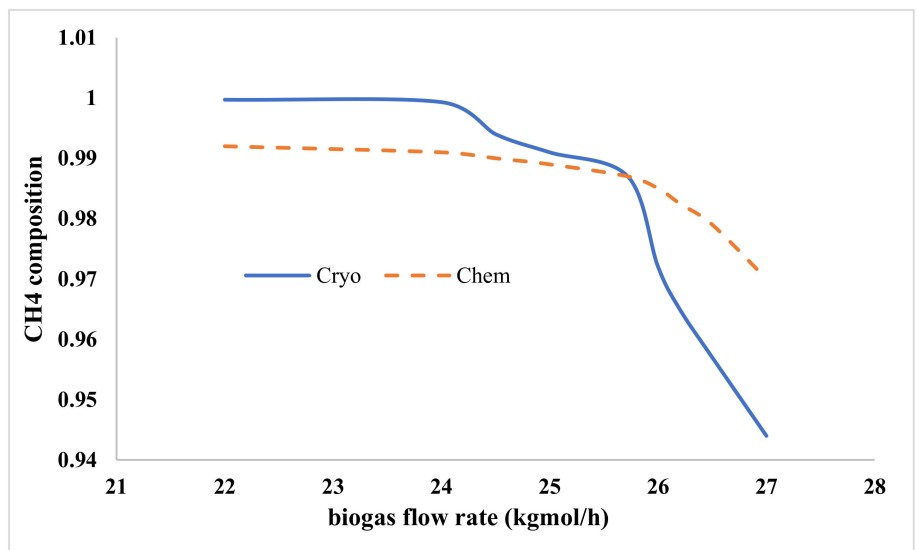

**Figure 9.** Effect of biogas flow rate on the purity of the methane output in each process.

*3.3. Energy Analysis*

The energy consumption for cryogenic separation and chemical scrubbing with the MEA solvent process equipment are presented in Tables 4 and 5, respectively. Moreover, LHV is the thermal value defined in $\frac{kj}{kg}$ and obtained from the simulator. The heat values are 18,661 $\frac{kj}{kg}$ for the input biogas and 48,358 $\frac{kj}{kg}$ and 48,317 $\frac{kj}{kg}$ for the biomethane produced via cryogenic separation and chemical scrubbing, respectively, as determined by the simulator's properties computations.

**Table 4.** Duties and power of the cryogenic separation process equipment.

| Equipment | Power Consumption/Duty (KW) |
| --- | --- |
| Compressor k-100 | 35.53 |
| Compressor k-101 | 34.67 |
| Compressor k-102 | 31.84 |
| Compressor k-103 | 145.2 |
| Reboiler | 67.19 |
| E-101 | 36 |
| E-102 | 46.39 |
| E-103 | 36.36 |
| E-104 | 64.01 |
| E-105 | 209.2 |
| Partial condenser | 49.8 |

*3.4. Economic Analysis*

The APEA economic tool was used for economic analysis. As a result, this tool was used to estimate capital and energy costs in the current research. The energy supply cost for the chemical absorption process includes coolant water ($\frac{MMgal}{h}$), electricity (KW), and steam ($\frac{Klb}{h}$) for reboiler heating (in saturated conditions and 100 psi pressure). Meanwhile, the supply cost of the cryogenic separation process includes electricity, coolant water, and refrigerant ($\frac{Klb}{h}$). Table 6 highlights the initial equipment costs for both methods. The

economic analysis of cryogenic separation and chemical scrubbing processes for biogas upgrading is presented in Table 7.

**Table 5.** Duties and power of the chemical scrubbing process equipment.

| Equipment | Power Consumption/Duty (KW) |
|---|---|
| Pump p-100 | 0.4943 |
| Pump p-101 | 0.182 |
| Reboiler | 401.5 |
| Partial condenser | 140.6 |
| E-100 | 7.135 |
| E-101 | 359.8 |
| E-102 | 245.4 |

**Table 6.** Prime costs of equipment for both methods.

| Chemical Absorption | | Cryogenic Separation | |
|---|---|---|---|
| Equipment | Price (USD) | Equipment | Price (USD) |
| E-103 | 60,800 | P-101 | 33,600 |
| E-102 | 65,800 | E-101 | 117,900 |
| E-104 | 81,200 | V-100 | 96,000 |
| Reboiler | 67,000 | T-100 | 199,100 |
| K-101 | 836,300 | P-100 | 33,500 |
| K-103 | 1,292,500 | T-101 | 341,000 |
| E-101 | 61,700 | E-100 | 56,400 |
| K-100 | 851,500 | E-103 | 59,000 |
| E-105 | 63,300 | E-102 | 70,100 |
| K-102 | 870,300 | P-100 | 33,500 |
| Condenser_@T-100 | 123,000 | | |
| Reboiler_@T-100 | 49,200 | | |
| Main Column_@T-100 | 160,800 | | |

**Table 7.** Results of economic computations.

| | Cryogenic Separation | Chemical Absorption |
|---|---|---|
| Energy cost (USD per year) | 252,407 | 185,937 |
| Investment cost (USD) | 10,303,200 | 4,481,080 |
| Equipment price (USD) | 3,471,800 | 218,400 |
| Cost of equipment with installation (USD) | 4,583,400 | 1,006,600 |
| Total annual cost (USD) | 3,686,807 | 1,679,630.33 |

The costs of cryogenic separation and chemical absorption of biogas are compared in Figure 10. As can be seen, the chemical absorption costs are significantly lower than the cryogenic method. The energy, capital, and total annual costs of chemical absorption are 26.33%, 56.51%, and 54.44% less than those of cryogenic separation, respectively. In terms of equipment procurement and installation, chemical absorption is also a cost-effective strategy. The initial cost of equipment for the chemical absorption approach is 78.3 percent cheaper than the initial cost for biogas cryogenic separation, according to the analysis.

Previous studies showed that the cryogenic process is more expensive than the chemical scrubbing process [45]. The difference between the initial investment cost and the operating cost of the two processes sometimes reaches more than 2.2 and 2.5 times, based on 600 $\frac{m^3}{h}$ (Approximately 5% difference in flow rate with this study) [46,47].

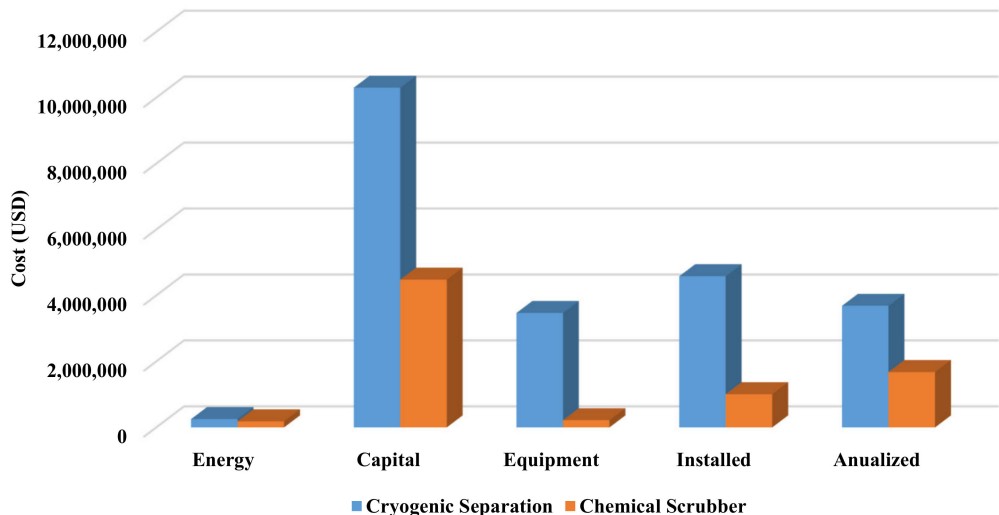

**Figure 10.** Cost comparison of cryogenic and chemical scrubbing of biogas.

*3.5. Exergy Analysis*

The values of the properties presented by the simulator were applied in the physical exergy computations, while the standard chemical exergy of methane (836,510 $\frac{kj}{kmole}$), carbon dioxide (275,430 $\frac{kj}{kmole}$), and water (3120 $\frac{kj}{kmole}$ liquid phase and 11,710 $\frac{kj}{kmole}$ steam phase) was cited in the study conducted by Kotas [48]. Regarding the lack of a standard chemical exergy value for MEA in Kotas et al.'s paper, this value was obtained from the research conducted by Ferrara et al. [49], equaling 1,975,173 $\frac{kj}{kmole}$. Regarding the issues above, the exergy values for cryogenic separation and chemical absorption processes were computed, and the results are presented in Table 8.

**Table 8.** Exergy loss and exergy efficiency for both methods.

| Equipment | $E_i^{in}(KW)$ | $E_i^{out}(KW)$ | $E_i^{loss}(KW)$ | $\eta_{Exergy}^i \cdot$ |
|---|---|---|---|---|
| | | cryogenic separation | | |
| K-100 | 36.17 | 29.78 | 6.39 | 0.8233 |
| E-101 | 30.86 | 24.05 | 6.81 | 0.7791 |
| K-101 | 57.94 | 51.69 | 6.25 | 0.8922 |
| E-103 | 52.81 | 45.97 | 6.81 | 0.8709 |
| K-102 | 77.86 | 71.97 | 5.89 | 0.9243 |
| E-102 | 118.47 | 107.22 | 11.25 | 0.9052 |
| E-104 | 111.75 | 97.69 | 14.06 | 0.8743 |
| K-103 | 159.53 | 131.81 | 27.72 | 0.8262 |
| E-105 | 138.17 | 115.14 | 23.03 | 0.8333 |
| T-100 | 89.78 | 71.11 | 18.67 | 0.7921 |
| Process | 873.43 | 746.43 | 127 | 0.85 |
| | | chemical absorption | | |
| T-10 | 2393.63 | 23,216 | 719.63 | 0.966 |
| E-100 | 0.53 | 0.52 | 0.01 | 0.981 |
| V-100 | 0.434 | 0.42 | 0.014 | 0.968 |
| E-102 | 27.07 | 5.24 | 21.83 | 0.194 |
| P-100 | 12.26 | 12.04 | 0.22 | 0.982 |
| E-101 | 99.95 | 84.75 | 15.2 | 0.85 |
| P-101 | 87.94 | 87.9 | 0.04 | 0.99 |
| T-101 | 27,744.52 | 20,275.74 | 7468.78 | 0.731 |
| Process | 51,908 | 43,682.61 | 8225.72 | 0.84 |

Figures 11 and 12 compare the equipment's exergy loss-share for the cryogenic separation and chemical absorption methods. As can be seen, the K-103 compressor (22%) in

the cryogenic separation method and the solvent recovery column (91%) in the chemical scrubbing method demonstrate the highest exergy losses. The K-103 compressor is used in the biogas refrigeration cycle before the distillation column to compress the refrigerant. One method of reducing exergy loss in the compressor is to employ a different refrigerant that can provide the biogas-required refrigeration at a lower working pressure. According to the analysis results, the total exergy loss in the cryogenic separation and chemical absorption methods is 127 KW and 8225.72 KW, respectively. The results indicate that the cryogenic separation method has a 98.46% lower exergy loss. According to, the most exergy loss in the chemical absorption process occurs in the T-101 solvent recovery column. Besides, the exergy efficiency of the equipment used in both methods was compared in Figures 9 and 10. As seen in Figures 13 and 14, in the cryogenic separation and chemical scrubbing methods, the K-102 compressor and the P-101 pump produced the highest exergy efficiency, respectively.

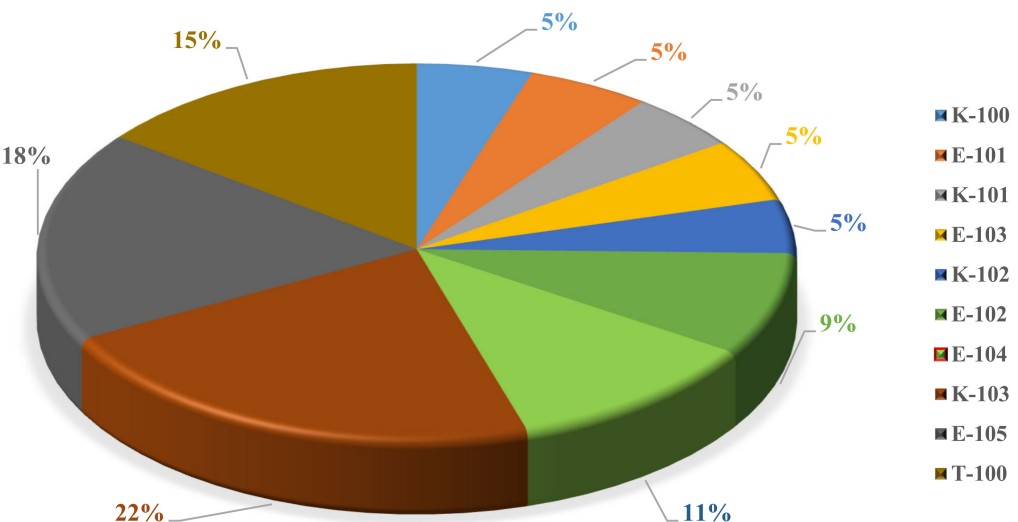

**Figure 11.** Comparison of exergy loss-share in the cryogenic separation method.

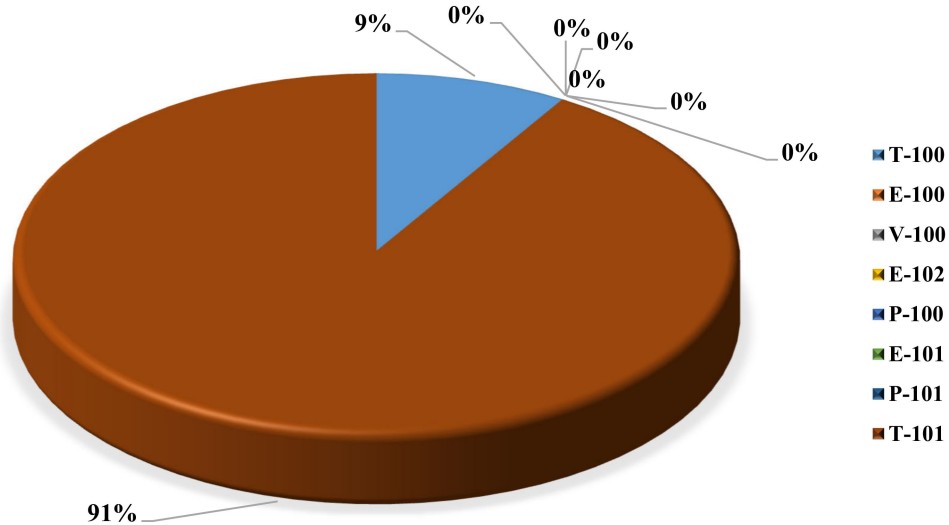

**Figure 12.** Comparison of exergy loss-share in the chemical absorption method.

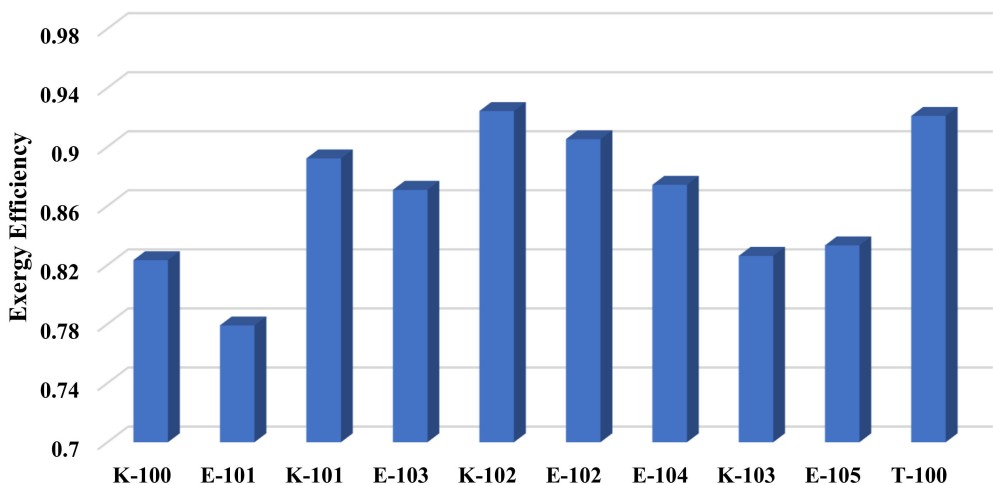

**Figure 13.** Comparison of the exergy efficiency of the equipment in the cryogenic separation method.

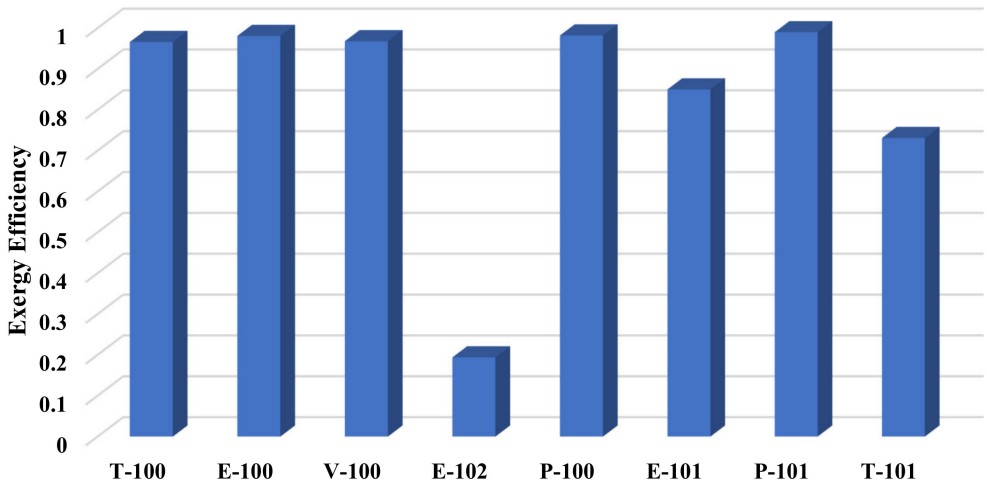

**Figure 14.** Comparison of the exergy efficiency of the equipment in the chemical absorption method.

### 3.6. Process Comparison

Table 9 summarizes key technological and economic discoveries in cryogenic separation and chemical absorption. The cryogenic separation approach surpassed the chemical absorption method in terms of thermodynamic performance, but the chemical absorption method is more economically feasible owing to reduced yearly expenditures. Naturally, the thermodynamic results are not significantly different, as the difference in the exergy domain is just 1%. This distinction enables us to group these two methods. However, in light of the importance of economic debates, we can ultimately introduce the chemical absorption method as a viable alternative.

**Table 9.** Comparison of technical and economic results.

| Parameter | Chemical Absorption | Cryogenic Separation |
|---|---|---|
| Total energy efficiency (%) | 89.85 | 91.92 |
| Total exergy efficiency (%) | 84 | 85 |
| Total annual cost ($) | 167,930.33 | 3,686,807 |

Vilardi et al. [18] simulated the chemical scrubbing process for biogas upgrading using the Aspen Plus software The results of the exergy analysis of their work showed that the

chemical scrubbing process has an exergy efficiency of 91.1%, which shows a good accuracy with the current research (89.85%).

Hashemi et al. [16] compared the processes of cryogenic distillation and chemical absorption and found that the process of cryogenic distillation had an 8% higher energy efficiency. More energy efficiency of the cryogenic process was observed, although the difference is lower (2.3%). This is due to two key factors. First, the distillation column was modeled as a single column in this research. While two-stage distillation improves energy efficiency somewhat, it also raises investment costs. The second explanation is that according to the kind of feed used in this research, the quantity of methane in biogas was greater. Moreover, some previous research showed that the cryogenic distillation process is more expensive than the chemical scrubbing process, and this difference is 1.43.7 times depending on the capacity [46,47,50].

### 3.7. Hydrogen Production

This study aimed to develop a method to produce hydrogen via biomethane reforming. Following a thermoeconomic analysis of cryogenic separation and chemical absorption, it was determined that chemical absorption was the most appropriate method for this purpose. A biomethane condensation compressor, heat exchangers, steam reformer reactors, and a hydrogen separation and gas-to-water conversion unit are all used in the process of reforming biomethane to create hydrogen. Figure 15 shows a schematic representation of the procedure. The following assumptions were considered in the biomethane reforming unit:

- The pressure of the reformer reactor is 1013 kPa [51].
- The temperature of the reformer and low-temperature gas–water conversion reactors are 700 °C and 200 °C, respectively, at a constant pressure of 1013 kPa [51].
- The adiabatic efficiency of the biomethane compressor is 75% [51].
- The pressure is constant throughout the entire hydrogen production process.
- The cooling temperature is 35 °C in the separator, so that the maximum separation of the synthesis gas is realized.
- The steam temperature to be mixed with biomethane equals 500 °C, and the temperature of biomethane to be mixed with steam is 400 °C [51].
- Hydrogen separation is completed in the PSA unit.
- The equilibrium model was used to simulate biomethane and water–gas reforming reactors, and the conversion rate was calculated using constant equilibrium data.

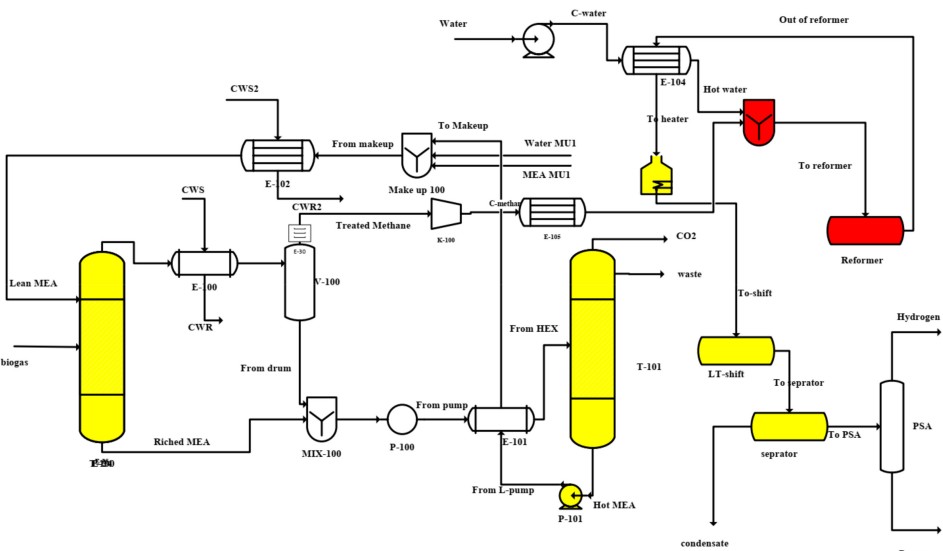

**Figure 15.** Schematic of the biomethane steam reforming process obtained from biogas chemical scrubbing.

Table 10 shows the thermodynamic and mass balances of the biomethane reforming process. The carbon monoxide and water steam reactions were carried out completely in the low-temperature gas–water conversion reactor since the steam reformer-leaving gas temperature reached 110.4 °C after a heat exchange with water. Table 11 summarizes the results, including the conversion rate in the reactors and the product generation and energy consumption indices.

**Table 10.** Composition and operational terms of flows in biomethane reforming.

| Stream | $CH_4$ | $CO_2$ | CO | $H_2O$ | T (°C) | P (kPa) | F ($\frac{kmole}{h}$) |
|---|---|---|---|---|---|---|---|
| Treated methane | 0.9873 | 0.0127 | 0 | 0.0224 | 12 | 105 | 16.15 |
| Water | 0 | 0 | 0 | 1 | 25 | 101.3 | 100 |
| C-water | 0 | 0 | 0 | 1 | 25.08 | 1013 | 100 |
| Hot water | 0 | 0 | 0 | 1 | 500 | 1013 | 100 |
| Out of reformer | 0.0225 | 0.0034 | 0.0895 | 0.6159 | 700 | 1013 | 142.3 |
| To reformer | 0.1365 | 0.0041 | 0 | 0.8594 | 479.4 | 1013 | 116.8 |
| C-methane | 0.9873 | 0.0127 | 0 | 0.0224 | 251.1 | 1013 | 16.15 |
| To heater | 0.0225 | 0.0034 | 0.0895 | 0.6159 | 110.4 | 1013 | 142.3 |
| To shift | 0.0225 | 0.0034 | 0.0895 | 0.6159 | 200 | 1013 | 142.3 |
| To separator | 0.0225 | 0.0927 | 0.0003 | 0.5266 | 200 | 1013 | 142.3 |
| To PSA | 0.0473 | 0.1940 | 0.0006 | 0.0059 | 35 | 1013 | 67.69 |
| Purge | 0.1909 | 0.7830 | 0.0024 | 0.0237 | 35 | 101.3 | 16.77 |
| condensate | 0 | 0.0007 | 0 | 0.9993 | 35 | 1013 | 74.59 |
| Hydrogen | | | | | 35 | 1013 | 50.92 |

**Table 11.** General results of hydrogen production through biomethane reforming.

| Parameter | Value |
|---|---|
| Methane conversion value in steam reformer (%) | 79.92% |
| Carbon monoxide conversion value in LT-Shift reactor (%) | 99.69% |
| $\theta_{H_2}\left(\frac{kmole_{H_2}}{kmole_{biogas}}\right)$ | 1.98 |
| $\beta_{H_2}\left(\frac{MJ}{kg_{H_2}}\right)$ | 90.48 |

In this section, several parameters such as the product generation intensity ($\theta_{H_2}$) and the energy consumption intensity for hydrogen production ($\beta_{H_2}$) are defined in Equations (28) and (29):

$$\theta_{H_2}\left(\frac{kmole_{H_2}}{kmole_{biogas}}\right) = \frac{\dot{G}_{H_2}}{\dot{G}_{biogas}} \tag{28}$$

$$\beta_{H_2}\left(\frac{MJ}{kg_{H_2}}\right) = \frac{Energy_{inlet}}{\dot{m}_{H_2}} \tag{29}$$

where $\dot{G}_{H_2}$ and $\dot{G}_{biogas}$ denote the molar flux of the produced hydrogen gas and biogas feed in $\frac{kmole}{h}$, respectively. Similarly, $Energy_{inlet}$ denotes the total energy consumed during the hydrogen production process. Its value (9292 $\frac{MJ}{h}$) is the sum of the energy consumed in the chemical absorption phase by the compressor, pumps, biomethane steam reformer, and solvent recovery column reboiler.

The specification of the methane stream from upgrading processes in cryogenic separation and chemical scrubbing is presented in Table 12. As can be seen from Table 12, the output of the cryogenic separation process has a higher temperature and pressure. Therefore, it has more ideal conditions compared to the product of chemical scrubbing process for reforming and hydrogen production. However, because the process is more expensive, especially the investment, the chemical scrubbing process is used. Figure 16 shows the utilities and power consumption or production in the process of methane reforming and hy-

drogen production. The cryogenic separation process has greater investment requirements, and therefore the superiority of the process' methane conditions might be neglected.

**Table 12.** Product methane stream profile in both processes.

| Stream | T ( °C) | P (kPa) | F ($\frac{kmole}{h}$) | % CH4 | % CO$_2$ |
|---|---|---|---|---|---|
| E-methane (cryo) | 142.7 | 3460 | 16.15 | 0.9874 | 0.0126 |
| Treated metane (chem) | 12 | 105 | 16.15 | 0.9873 | 0.0127 |

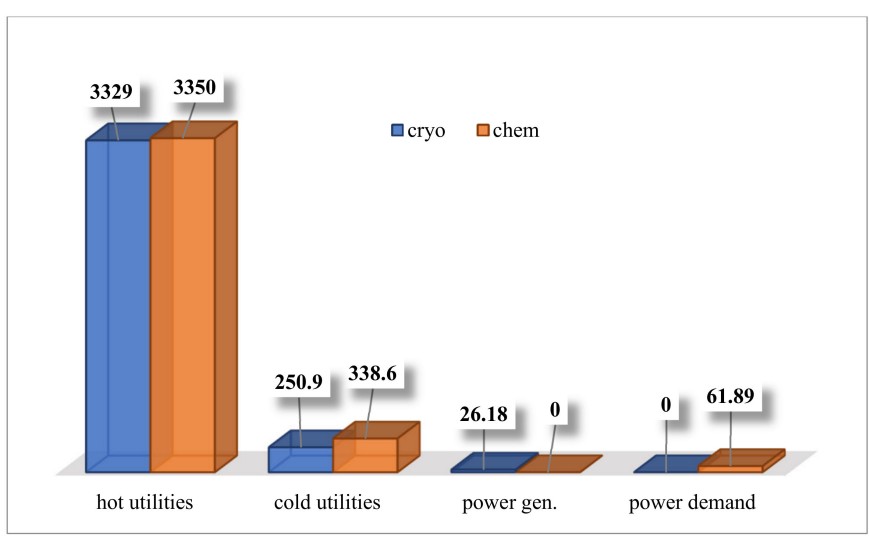

**Figure 16.** Comparison of utility and power consumption or production in methane-reforming process.

The current study simulates and compares two important processes in biogas upgrading with static simulation. It is suggested that to complete the evaluation of the processes, as well as study the process challenges in the field of exergy, we need advanced exergy studies on the simulation of biogas upgrading processes with economic parameters in mind. Furthermore, the low concentration of hydrogen sulfide and other biogas compounds (which have a small amount) is omitted, which can be completed in future studies, taking into account these factors, as well as an optimization based on the number of stages, column, operating conditions, number of stages of pressurization, distillation columns, etc.

## 4. Conclusions

This study simulated two cryogenic separation and chemical absorption processes for biogas upgrading and biomethane production and conducted a 3E (Energy, Exergy, and Economic) analysis. Based on the economic analysis results, the annual cost and investment cost of cryogenic separation is 2.2 and 2.3 times higher. The chemical scrubbing and cryogenic separation methods have a total exergy efficiency of 84% and 85%, respectively. Naturally, when the thermodynamics of the loss is considered, the cryogenic separation method demonstrates a significantly superior performance, with an exergy loss 98.46% lower than that of the chemical scrubbing method. As a consequence, the cryogenic approach outperforms chemical cleaning in terms of thermodynamic performance. The economics, on the other hand, influences the ultimate decision, making chemical cleansing more appealing and cost-effective. Comparing the results indicated that chemical absorption was the more effective method for upgrading biogas, which was used to produce hydrogen via biomethane steam reforming. The results indicated that this method consumed 90.48 MJ of energy per kilogram of hydrogen produced and produced hydrogen at a rate of 1.98 mol per mole of biogas (68% methane and 32% carbon dioxide).

**Author Contributions:** Conceptualization, E.N., A.N. and H.G.; methodology, E.N., A.N. and H.G.; software, E.N.; validation, E.N. and A.N.; formal analysis, E.N. and H.G.; investigation, E.N., A.N.

and H.G.; resources, A.N.; data curation, A.N. and H.G.; writing—original draft preparation, E.N.; writing—review and editing, E.N., A.N. and H.G.; visualization, A.N., H.G.; supervision, A.N. and H.G.; project administration, A.N. All authors have read and agreed to the published version of the manuscript.

**Funding:** This research received no external funding.

**Institutional Review Board Statement:** Not applicable.

**Informed Consent Statement:** Not applicable.

**Data Availability Statement:** Data will be available upon request.

**Conflicts of Interest:** The authors declare no conflict of interest.

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
