# Peer review of "Energy, Exergy, and Economic Analysis of Cryogenic Distillation and Chemical Scrubbing for Biogas Upgrading and Hydrogen Production"

_sustainability, doi:10.3390/su14063686_

Round 1

Reviewer 1 Report

General comments:

The manuscript is about the study of 3E analysis of cryogenic and chemical biogas up-gradation and hydrogen production. This study showed that chemical scrubbing for biogas upgrading was preferable method in comparison to economics of cryogenic distillation though there was not much difference in energy and exergy analysis. The study is interesting and relevant with the scopes of the journal. The methodology part could be more concise and some of the sections of the results could be moved to methodology. There are some typos which need to be corrected.

Specific comments are given below with line number:

Line 37-38: The sentence “Similarity, the Copenhagen……..2100” is not clear which needs to be rephrased. Similar comments for line 57-58.

Line 118: Need to mention model version, company and country of origin of Aspen HYSYS Software

Line 221-227: May consider moving this part to the methodology

Line 237, 238: Table 2 and Table 3 are not clear where these are input variables for the simulation software or results which need to be clarified.

Line 262: correct spelling “separation”.

Line 335 to 336: This sentence “The total exergy loss ……exergy loss is 98.46%” is not clear. The values in the sentence have written with comma and dot which should be unified.

Figure 12 and 13 could be deleted since the values of exergy efficiency were given in Table 8.

Author Response

Thanks for reviewing the manuscript
Please refer to the attachment
Sincerely Yours

Reviewer 2 Report

The article reports the biogas upgradation. The work seems interesting but there are some major flaws to overcome in the revision.

  1. The novelty and contribution are missing in the introduction and the most recent literature is not considered.
  2. 2. Fig 5 need to be rechecked for mistakes. 
  3.  The discussion-based on results is missing fr Fig 5-6
  4. The results can be analysed in a way better than in the current form. The organisation of figures can be better.
  5. Authors must include the latest literature in the discussion and compare it with the literature. 
  6. The conclusion can be reduced. 
  7. Various typos can be seen in the manuscript.

Author Response

Thanks for reviewing the manuscript
Please refer to the attachment
Yours sincerely

Reviewer 3 Report

I read carefully an interesting and comprehensive research work entitled Energy, exergy and economic analysis of cryogenic distillation and chemical scrubbing for biogas upgrading and hydrogen production. The concept of the article is interesting and suitable for Sustainability. This manuscript is generally well written and clearly presented however still needs to address some comments, and thus require  revision to improve the quality of the manuscript.

  1. Title should modify which can describe the whole research work.
  2. A well addressed graphical scheme of study design should be inserted.  
  3. In the introduction section, write the novelty of the work and the problem statement clearly.
  4. Techno Economic challenges of the developed system and future research directions that need to be described by adding a new section before the conclusions section?
  5. The conclusion of the study needs to be added with the specific output obtained from the study, it could be modified with precise outcomes with a take home message.
  6. Some English and grammar mistakes are present that need to be correct to improve the quality of the manuscript.

Author Response

thanks for reviewing manuscript
Please refer to the attachment
Yours sincerely
